# Enhancement of Piezoelectric Properties of Flexible Nanofibrous Membranes by Hierarchical Structures and Nanoparticles

**DOI:** 10.3390/polym14204268

**Published:** 2022-10-11

**Authors:** Feng Wang, Hao Dou, Cheng You, Jin Yang, Wei Fan

**Affiliations:** 1School of Textile Science and Engineering, Xi’an Polytechnic University, Xi’an 710048, China; 2State Key Laboratory of Intelligent Textile Material and Products, Xi’an Polytechnic University, Xi’an 710048, China

**Keywords:** piezoelectric property, polyacrylonitrile, poly(vinylidene fluoride), ZnO nanorods, electrospinning

## Abstract

Piezoelectric nanogenerators (PENGs) show superiority in self-powered energy converters and wearable electronics. However, the low power output and ineffective transformation of mechanical energy into electric energy l limit the role of PENGs in energy conversion and storage devices, especially in fiber-based wearable electronics. Here, a PAN-PVDF/ZnO PENG with a hierarchical structure was designed through electrospinning and a hydrothermal reaction. Compared with other polymer nanofibers, the PAN-PVDF/ZnO nanocomposites not only showed two distinctive diameter distributions of uniform nanofibers, but also the complete coverage and embedment of ZnO nanorods, which brought about major improvements in both mechanical and piezoelectric properties. Additionally, a simple but effective method to integrate the inorganic nanoparticles into different polymers and regulate the hierarchical structure by altering the types of polymers, concentrations of spinning solutions, and growth conditions of nanoparticles is presented. Further, the designed P-PVDF/ZnO PENG was demonstrated as an energy generator to successfully power nine commercial LEDs. Thus, this approach reveals the critical role of hierarchical structures and processing technology in the development of high-performance piezoelectric nanomaterials.

## 1. Introduction

With the coming era of smart wearable devices, there is an increasing need for self-powered wearable electronics, and the advancement of wearable electronics is largely dependent on the power output of power sources [1,2]. The occurrence of piezoelectric nanogenerators (PENGs) has solved the long-standing problem of energy sources for wearable devices [3]. As essential components of PENGs, piezoelectric polymers can effectively convert mechanical energy into electric energy [4]. In general, piezoelectric polymers possess reasonable piezoelectricity, high mechanical flexibility, sensitivity to voltage change, and low impedance, and have wide applications in gas, liquid, and biological sensors [5]. According to their physical structure, piezoelectric polymers can be classified into semicrystalline or amorphous polymers [6]. Poly(vinylidene fluoride) (PVDF) is a well-known semicrystalline piezoelectric polymer with strong piezoelectricity in its stretched and polarized forms. Many studies have shown that PVDF nanofibrous membranes with piezoelectric output can be obtained by electrospinning [7,8]. In addition, PAN, as a commonly used amorphous polymer, has demonstrated excellent piezoelectric properties. Wang et al. found that electrospun PAN nanofibers can generate stronger piezoelectricity even than PVDF nanofibers due to their high content of planar Sawtooth PAN conformation [9]. Unfortunately, it is still a challenge to further substantially improve the piezoelectricity of polymer-based nanomaterials.

In order to solve the above issue, some researchers who are interested in the micro/nanostructured design of piezoelectric composite materials, such as porous membranes [10], nanopillar arrays [11], and trigonal line-shaped and pyramid-shaped membranes [12], aim to simultaneously obtain high electrical performance and flexibility in nanocomposites. For example, Mu et al. used two piezoelectric polymers to achieve a much higher acoustoelectric than their individual counterpart by increasing the density and generating a synergistic piezoelectric effect [13]. Hence, the fabrication of hybrid micro-nanofibers with excellent piezoelectric properties and large differences in fiber diameters by selecting two polymers is an essential method.

Another research direction in this field is focused on adding particle doping onto piezoelectric polymers, which can utilize various properties of inorganic nanomaterials to improve the piezoelectric resistance of nanofibrous PENGs. The most commonly used nanoparticles include ZnO [14], ZrO [15], graphene oxide (GO) [16], Al_2_O_3_ [17], TiO_2_ [18], carbon nanotubes [19], and others. Among these, ZnO is a kind of semiconductor material, and the asymmetric central crystal of ZnO and its structure give it piezoelectric properties. Singh et al. reported that ZnO nanorods can significantly enhance the piezoelectric properties of PVDF nanofibrous membranes [20]. Sun et al. incorporated ZnO nanorods into PAN nanofibers with improvements in energy-harvesting ability of about 2.7 times [5]. It is obvious that enriching hierarchical structures by integrating organic-inorganic materials is an effective and simple approach to increasing the piezoelectric properties of PENGs to a large extent.

In this paper, PAN-PAN (P-PAN), PAN-PVDF (P-PVDF), and PAN-PVA (P-PVA) nanofibrous membranes were prepared by changing the types of polymers or concentrations in order to regulate the hierarchical structure. After comparing the piezoelectric and physical properties, the P-PVDF nanofibrous membrane was selected as the optimal outcome for the in situ growth of ZnO nanorods. The P-PVDF/ZnO membrane shows higher piezoelectric properties and fracture stress based on the hierarchical structure, and can lighten nine commercial LEDs. Thus, this approach offers a new strategy for the enhancement of the piezoelectric properties of PENGs, with potential applications in flexible wearable electronics, health monitoring, and signal sensing. 

## 2. Materials and Methods

### 2.1. Materials

All chemicals used in the experiments were of analytical grade. PVDF (HSV900, Mw~1,000,000) was obtained from Arkema, France. PAN (number average molecular weight, Mw~15,000) was purchased from Shanghai Chenqi Chemical Technology Co., Ltd., Shanghai, China. PVA (degree of polymerization: 1700; degree of hydrolysis: 88%) was provided by Aladdin Biochemical Technology Co., Ltd., Shanghai, China. N, N-dimethylformamide (DMF), hexamethylenetetramine (C_6_H_12_N_4_), zinc nitrate hexahydrate (Zn(NO_3_)_2_·6H_2_O), isopropyl alcohol (IPA), and ammonia hydroxide (NH_3_·H_2_O) were purchased from Chengdu Colon Chemicals Co., Ltd., Chengdu, China. Zinc acetate (Zn(AC)_2_) was obtained from Chengdu Aikeda Chemical Reagent Co., Ltd., Chengdu, China. Acetone was obtained from Sinopharm Chemical Reagent Co., Ltd., Shanghai, China.

### 2.2. Preparation of Nanofibrous Membranes 

First, the electrospinning solutions were prepared by dissolving PAN (10% *w*/*v*), PAN (6% *w*/*v*) in DMF, PVA (10% *w*/*v*) in DI water, and PVDF (14% *w*/*v*) in the mixture of DMF and acetone (volume ratio 7:3). The solutions were stirred at room temperature for an appropriate time. Next, nanofibrous membranes were prepared with electrospinning equipment. The corresponding solutions were fed into a 10-milliliter plastic syringe controlled by a syringe pump. The process of producing nanofibers is shown in Figure 1a. A high voltage of 16 KV was applied and electrospinning of two spinning solutions simultaneously was performed at an appropriate flow rate from 0.2 mL·h^−1^ to 0.6 mL·h^−1^, with a spinning distance of 13 cm. Nanofibers were collected on a grounded collector for 3 h for the uniform thickness.

### 2.3. Preparation of P-PVDF/ZnO Nanofibrous Membrane

Zn (AC)_2_ was dissolved in IPA (50 mL) and stirred vigorously at 60 °C for 20 min to obtain a seed solution (10 mM). The nanofibrous membranes were immersed in the seed solution for 20 min and then cured at 80 °C for 30 min. This process was repeated three times. All samples prepared with the ZnO seed layer were used for the hydrothermal growth in the next process.

ZnO nanorod arrays were prepared by low-temperature hydrothermal synthesis technology. The standard growth solution consisted of Zn(NO_3_)_2_·6H_2_O (30 mM), C_6_H_12_N_4_ (10 mM), and NH_3_·H_2_O (5 mL) and deionized (DI) water (100 mL) with stirring at room temperature for 3 min. The mixed solution and P-PVDF nanofibrous membranes were transferred into a 100-milliliter stainless-steel autoclave and maintained at different temperatures (85, 95, and 105 °C) for 3 h. Finally, the samples with ZnO nanorods were rinsed with a large amount of running DI water, ultrasonically cleaned for 30 min to remove the residue, and then dried in air to obtain P-PVDF/ZnO nanofibrous membranes. ZnO growth solutions with five concentrations were prepared, based on the standard solution as 100% and corresponding composition dilutions from 80% to 20%, which were labeled as shown in Table 1. 

### 2.4. Fabrication of PENGs

Figure 1b shows the real photograph of the P-PVDF/ZnO PENG and schematically illustrates the structure of piezoelectric device and test setup. All nanofibrous membranes obtained were cut into small pieces with an effective working area of 40 × 40 mm^2^. Next, the silver fabric (30 × 30 mm^2^) used for electrodes was placed on both sides of the nanofibrous membranes, and conductive copper tapes were attached to these two sliver fabric electrodes to collect the output voltage signals. To protect from any external mechanical damage, the polyethylene terephthalate (PET) membrane was further covered on the PENGs to make an encapsulation using a laminator. 

### 2.5. Characterizations

The surface morphology of the nanofibrous membranes was characterized by scanning electron microscopy (SEM, Quanta-450-FEG + X-MAX50, FEI, Switzerland) and the elements of P-PVDF/ZnO were tested with energy-dispersive spectroscopy (EDS). It is worth noting that all samples should be sputter-coated with gold before surface-morphology observation. Fourier-transform infrared spectroscopy (FTIR) experiments were carried out in attenuated-total-reflectance (ATR) mode, and the spectra of nanofibrous membranes were recorded with a Fourier-transform spectrometer (Spotlight 400, Perkin Elmer, MA, USA) in the range of 650 to 4000 cm^−1^ with resolution of 4 cm^−1^. The composition and crystallographic structures were confirmed by X-ray diffraction (XRD, D8 Advance, Bruker, Germany) with Cu Kα radiation operated at 40 kV and 30 mA and the sample was scanned in the 2θ range of 15–70°. Mechanical properties of all membranes cut into 50 mm (length) × 10 mm (width) were measured by a universal testing device (UTM5205X, Shenzhen, China) with stretching rate of 10 mm/min at room temperature until the sample was broken. The final results were the average value of at least 5 measurements.

A self-built device which consisted of a piezoelectric device, a computer, a controller, and an electrochemistry workstation (Keithley 6514, Tektronix, OR, USA) was used for measuring the piezoelectric properties (as shown in Figure 1b). The real-time voltage output of the PENG with dimensions of 30 mm × 30 mm was recorded by the electrochemistry workstation and computer. During testing, the amplitude and frequency of compressive impacts were controlled by the controller. The PENG was repeatedly tapped through piezoelectric device at a fixed frequency of 2 Hz and amplitude of 10 mm.

## 3. Results and Discussion

### 3.1. Morphologies of Different Hierarchies of Nanofibers 

According to Figure 2a–c, different morphologies and sizes of electrospun nanofibers were clearly presented due to the distinctive polymer composition of the spinning solutions. All three nanofibers showed one thin part and one thick part of the diameter distribution, but different morphologies. In Figure 2a, two distinctive diameters of P-PAN nanofibers, whose average diameters were 120 nm and 230 nm, respectively, are shown, and discontinuous beads can be observed. The beads were caused by the low viscosity of the spinning liquid with low concentration. For the P-PVDF nanofibers in Figure 2b, a more obviously hierarchical structure appeared, with average diameters of 210 nm and 640 nm. In addition, all the nanofibers were smooth and uniform, indicating the integration of PAN and PVDF. Hierarchically structured P-PVA nanofibers with diameters 140 nm and 270 nm also occurred. Because the deionized water was the solvent of the PVA, adhesion between the nanofibers and the rough surface was observed. 

### 3.2. FTIR Analysis of Different Hierarchies of Nanofibers

Figure 2d presents the FTIR spectra of the three different nanofibers. The P-PAN membrane exhibited characteristic absorption peaks from the PAN near the wavenumbers of 1236 cm^−1^, 1375 cm^−1^, 1453 cm^−1^, 1731 cm^−1^, 2239 cm^−1^, and 2937 cm^−1^. Of these, the characteristic peaks near 1375 and 2937 cm^−1^ were related to the bending vibration and the contraction vibration of CH_2_, respectively; the characteristic peak at 2239 cm^−1^ corresponds to the rocking vibration of the cyan (C≡N); the characteristic absorption intensity near 1731 cm^−1^ was attributed to carbonyl (C=O) [21]. In addition, the characteristic peak at 1453 cm^−1^ was related to the in-plane bending of CH. The absorption peak at 1236 cm^−1^ corresponds to the 3^1^- helical conformation of the identical sequence in the PAN molecular structure [22,23]. For the P-PVA, approximately 3300 cm^−1^, 1421 cm^−1^, and 1092 cm^−1^ correspond to the stretching-vibration peak of hydroxyl (-OH), the bending-vibration peak of CH-OH stretching, and the stretching-vibration-absorption peak of the C-O stretching, respectively [24], which belongs to the characteristic peaks of the PVA. It can be clearly observed that the characteristic peak of the PVDF appeared in the FTIR spectra of the P-PVDF. The characteristic absorption peaks at 765 cm^−1^, 875 cm^−1^, 1075 cm^−1^, 1176 cm^−1^, and 1405 cm^−1^ were produced by the α crystal form of the PVDF, and the β crystal has prominent characteristic absorption peaks at 840 cm^−1^ [25]. Furthermore, the characteristic peaks of the PAN can be seen in all three curves, which indicates the co-existence of two polymers in the nanofibers by double-nozzle electrospinning, as in the results of the SEM.

### 3.3. Piezoelectric and Mechanical Properties of Different Hierarchical Nanofibers

The results from the piezoelectric test of the P-PAN, P-PVDF, and P-PVA nanofibrous membranes are shown in Figure 3a–c, respectively. It is worth noting that all nanofibrous membranes can generate voltage output, but the peak voltage of the P-PVDF PENG can reach 7.2 V, mainly due to the molecular structure of PVDF β crystal phase [26]. The peak voltage of the P-PAN PENG was 3.4 V, about half of that of the P-PVDF PENG, because PAN owns the high dipole moment and 3^1^-helical [27,28]. Further, the output voltage of the P-PVA PENG was only 1.2 V due to the low piezoelectricity of the PVA. 

From the tensile-test curves of the three nanofibrous membranes shown in Figure 3d, it can be observed that the fracture stress of the P-PVA and P-PAN nanofibrous membranes was nearly the same, at 3.4 MPa and 3.5 MPa, respectively, while the P-PVDF nanofibrous membrane reached 4.3 MPa. This can be attributed to the excellent uniformity, fineness, and lack of defects of P-PVDF nanofibers. The above results show that the piezoelectric output and mechanical properties of P-PVDF are optimal. Therefore, the P-PVDF was selected to explore the further enhancement of piezoelectric properties through the in situ growth of ZnO nanorods.

### 3.4. Morphologies of P-PVDF/ZnO Nanofibrous Membranes

Figure 4 shows the SEM image of the P-PVDF/ZnO generated by the ZnO growth solutions with five different concentrations during the hydrothermal reaction. It can be easily observed that the ZnO-growth-solution concentration had a critical effect on the coating and morphology of the ZnO. In the 100%-growth solution in Figure 4a, the ZnO was nanorod-shaped and perfectly embedded into the P-PVDF nanofibers. When the concentration was 80%, the nanorod shape became a nanoneedled structure and the amount of ZnO decreased, as shown in Figure 4b. The reduction in ZnO-growth-solution concentration to 60% led to the scattering of a few nanoclusters over the nanofibers, which can be seen in Figure 4c. Furthermore, no obvious nanorods grew on the surfaces of the P-PVDF nanofibers shown in Figure 4d–e, resulting from the low concentration of the ZnO growth solution. Furthermore, the element distribution of the P-PVDF/ZnO nanofibers was verified via EDS from a single nanofiber (Figure 4f), which demonstrates the successful growth of the ZnO on the P-PVDF nanofibers.

On the other hand, the growth temperature of ZnO also plays an essential role in the formation of hierarchically structured nanofibers. Figure 5a–c shows the difference in the morphologies of the P-PVDF/ZnO nanofibers at 100% ZnO growth solution with temperatures of 85 °C, 95 °C, and 105 °C. A few nanoneedled structures can be seen at 85 °C, because the temperature was too low to favor the growth of ZnO nanorods. However, after the temperature was increased to 95 °C, a large quantity of ZnO nanorods completely coated the nanofibers, tightly forming the nanocomposite with the P-PVDF nanofibers. When the temperature reached 105 °C, the coverage of the ZnO nanorods dramatically dropped, and some smooth nanofibers were exposed. Hence, this is a simple way to further regulate the hierarchical structure of P-PVDF/ZnO nanofibers by controlling the growth temperature.

### 3.5. XRD Analysis of P-PVDF/ZnO Nanofibers

The XRD patterns prove the difference in the molecular conformation of the PAN, PVDF, P-PVDF/ZnO nanofibers, and ZnO powder. As shown in Figure 5d, the diffraction peaks at 2θ = 31.7, 34.4, 36.2, 47.5, 56.6, 62.8, 66.1, 67.9, and 69.0° correspond to the (100), (002), (101), (102), (110), (103), (200), (112), and (201) crystal faces of the ZnO hexagonal wurtzite structure, respectively [29]. The diffraction peaks at 2θ = 18.4 and 26.7° correspond to the crystallization peaks of the PVDF [30]. As suggested by the XRD pattern of the PVDF nanofiber, the diffraction peak around 20.5° can be attributed to the (110) and (200) planes of the β-phase content of the PVDF [31]. The XRD spectrum of the PAN consisted of characteristic peaks at 2θ = 16.8 and 17.0° [27]. The diffraction peaks of the PAN, PVDF, and ZnO powder appeared in the XRD patterns of the P-PVDF/ZnO. Hence, the XRD patterns also demonstrated the growth of the ZnO nanorods on the P-PVDF nanofibers.

### 3.6. Piezoelectric and Mechanical Properties of P-PVDF/ZnO Nanofibers

In order to further enhance the piezoelectric properties of the P-PVDF nanofibrous membrane, typical ZnO nanorods were hydrothermally grown on the surfaces of the P-PVDF nanofibers, since ZnO nanoparticles have a relative high surface activity. Figure 6a,b shows that the P-PVDF/ZnO PENG had the highest piezoelectric performance, yielding a peak voltage of 16.0 V. Compared with those of the PENGs without the ZnO nanorods, the output voltage of the P-PVDF/ZnO PENG was significantly increased, at least doubling. There are several reasons for this phenomenon. Firstly, ZnO nanorods can contribute to the formation of β-phase crystallization of PVDF and the zigzag conformation of PAN molecular chains [5,15], resulting in the enhancement of piezoelectric properties of P-PVDF/ZnO PENG. Secondly, P-PVDF/ZnO PENGs with hierarchical structures produce greater deformation under compressive force, resulting in a larger voltage output. Finally, PVDF, PAN, and ZnO have intrinsic piezoelectric properties, creating synergistic piezoelectric effects.

Furthermore, the mechanical properties were also improved by adding the ZnO nanorods. As shown in Figure 6c, the tensile strength of the P-PVDF/ZnO nanofibrous membrane was 20% higher than that of the P-PVDF nanofibrous membrane, but the breaking elongation of the P-PVDF/ZnO nanofibrous membrane was sharply reduced to about one sixth of that of the P-PVDF nanofibrous membrane, indicating the change from toughness to rigidity. This was due to the deformation of the molecules of the polymers at 95 °C during the hydrothermal reaction and the reinforcement of the ZnO nanorods fully embedded as nanofillers. Notably, the PENG could be used in energy-harvesting applications, such as flexible electronics, by transferring mechanical energy to electrical energy. As shown in Figure 6d, the energy generated under 2 Hz lasting pressure was usable and nine commercial LEDs were powered. The related video can be found in Appendix A.

## 4. Conclusions

In summary, hierarchically structured nanofibers can be easily fabricated by the use of two polymers in electrospinning and by in situ growth of nanoparticles in hydrothermal reactions. The SEM images clearly show the two-part diameter distribution of the nanofibers, as well as the full embedment of the ZnO nanorods as reinforcement fillers. The results of the FTIR and XRD confirmed every component in the nanocomposites. Furthermore, owing to the synergistic piezoelectric effect of organic and inorganic piezoelectric materials, namely the typical PVDF, the PAN polymers, and the hierarchically arranged ZnO nanorods, both the mechanical and the piezoelectric performance were significantly boosted, showing 5.25 MPa of tensile strength and 16.0 V of voltage output. Further, the designed PENG with the hierarchical structure was used for the energy harvesting, lighting nine commercial LEDs under 2 Hz of lasting pressure. Therefore, the high-performance flexible PENG can be potentially applied to wearable electronics. 

## Figures and Tables

**Figure 1 polymers-14-04268-f001:**
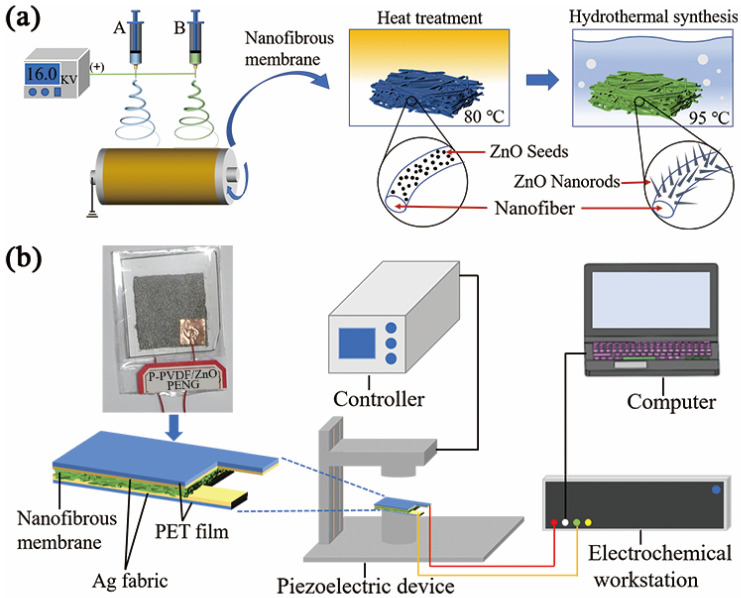
Schematic diagram of the experimental setup. (**a**) Nanofibers fabricated by double-nozzle electrospinning. (**b**) The real photograph of the P-PVDF/ZnO PENG and schematic illustration of the determination of piezoelectric properties.

**Figure 2 polymers-14-04268-f002:**
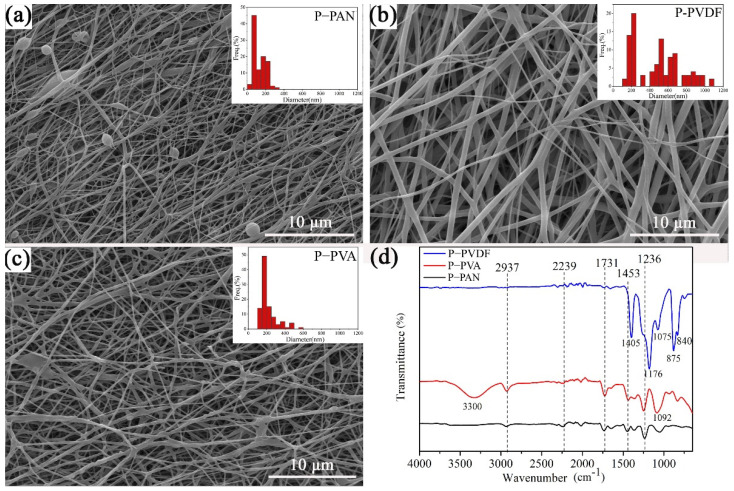
SEM and fiber-diameter distribution of (**a**) P-PAN, (**b**) P-PVDF, and (**c**) P-PVA and (**d**) FTIR spectra of different micro-nanofibers.

**Figure 3 polymers-14-04268-f003:**
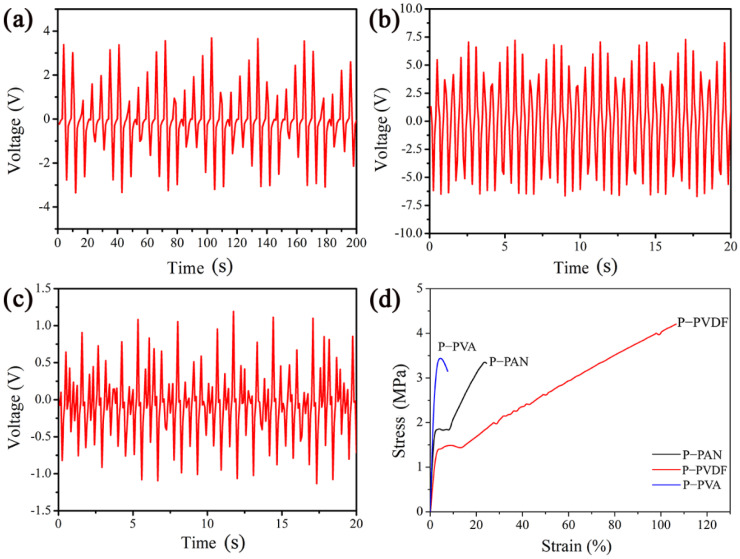
Output voltage generated by the nanofibrous membranes (**a**) P-PAN, (**b**) P-PVDF, and (**c**) P-PVA. (**d**) Comparison of tensile properties of P-PAN, P-PVDF, and P-PVA nanofibrous membranes.

**Figure 4 polymers-14-04268-f004:**
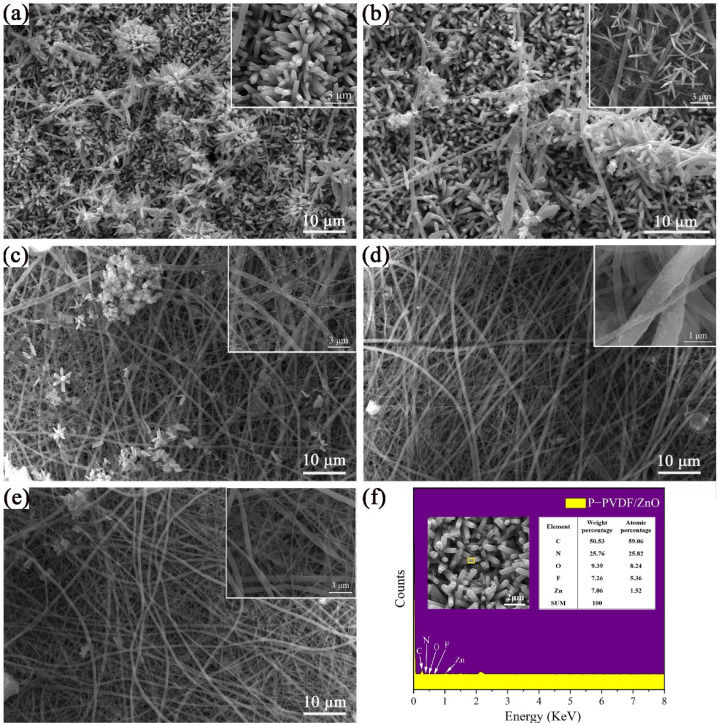
SEM of P-PVDF/ZnO nanofibrous membranes by hydrothermal reaction with different growth solutions: (**a**) 100%, (**b**) 80%, (**c**) 60%, (**d**) 40%, and (**e**) 20%. (**f**) The element distribution of P-PVDF/ZnO nanofibers via EDS.

**Figure 5 polymers-14-04268-f005:**
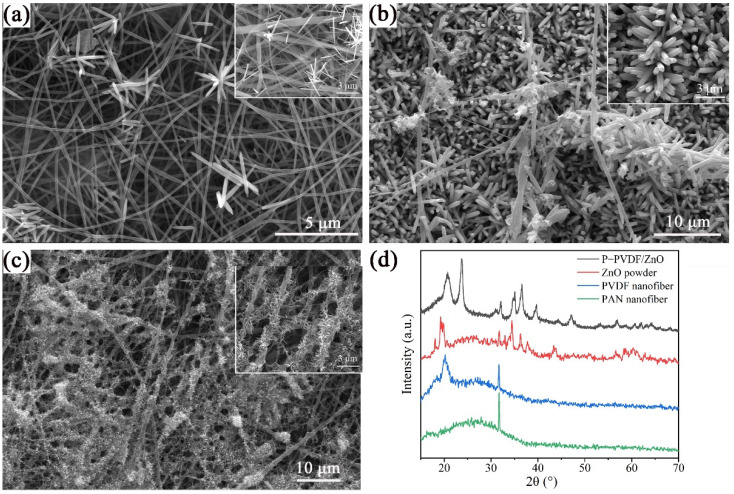
SEM of P-PVDF/ZnO nanofibers by hydrothermal reaction with different temperatures: (**a**) 85 °C, (**b**) 95 °C and (**c**) 105 °C. (**d**) XRD patterns of PAN, PVDF, and P-PVDF/ZnO nanofibers and ZnO powder.

**Figure 6 polymers-14-04268-f006:**
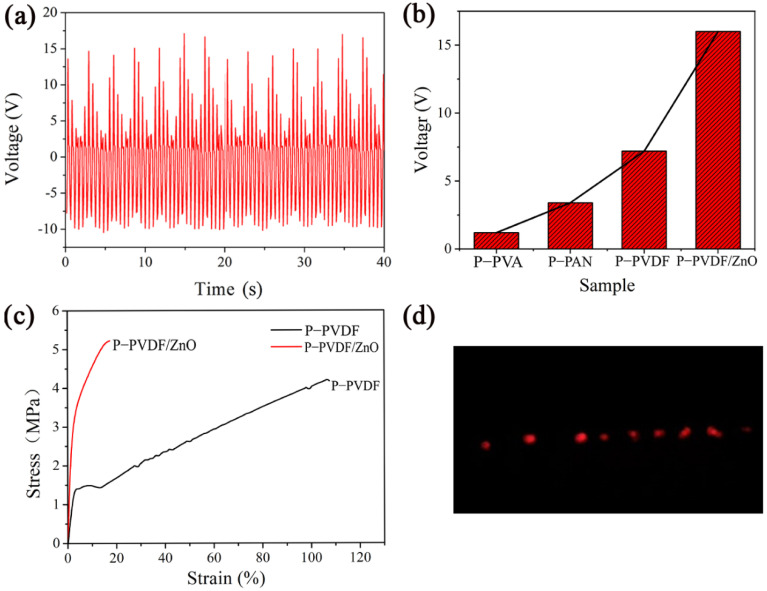
(**a**) Voltage output of P-PVDF/ZnO PENG. (**b**) Changes in voltage output with different PENGs. (**c**) Comparison of tensile stress between P-PVDF and P-PVDF/ZnO nanofibrous membranes. (**d**) Nine commercial LEDs were lit by tapping P-PVDF/ZnO PENG (working area, 9 cm^2^; impact frequency 2 Hz).

**Table 1 polymers-14-04268-t001:** The sample formulation of growth solutions with the different parameters.

Sample	Zn(NO_3_)_2_·6H_2_O (mM)	C_6_H_12_N_4_ (mM)	NH_3_·H_2_O (mL)	DI Water (mL)
100%	30	10	5	100
80%	24	8	4	100
60%	18	6	3	100
40%	12	4	2	100
20%	6	2	1	100

## Data Availability

Not applicable.

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
