# Peer review of "Enhancement of Piezoelectric Properties of Flexible Nanofibrous Membranes by Hierarchical Structures and Nanoparticles"

_polymers, 2022, doi:10.3390/polym14204268_

Round 1

Reviewer 1 Report

Dear Authors 

The article presented here reports the study of PAN-PVDF/ZnO nanofibers composites for piezoelectric applications. The article was very well written and structured, with the methods and results clearly and well described. Electrospinning produces beautiful nanofiber composites, and the piezoelectric device shows a great opportunity for the next development of electronic materials.

Reviewer 2 Report

The manuscript deals with PVDF-based nanofibrous membranes for applications in wearable electronics. The membranes consist of PVDF, PAN and ZnO nanorods, the latter being grown on the electrospun fibres in a hydrothermal process. There is no doubt that the topic is of high importance and interesting for many researchers. The manuscript is well organized, the experimental data is presented well and very convincing. Thus, the manuscript is suited for publication in Polymers.

Before publication the manuscript requires revision.

1. Firstly, the experimental details are missing. The suppliers of all analytical instruments used needs to be given. Moreover, information is missing how IR spectra were measured, e.g. AFM, photoacoustic cell, ….? The sample preparation is missing, too.

2. Further, details of the PVDF used are required, e.g. average molar masses or at least the melt flow index.

3. The description of the nanofiber membranes at the end of page 2 does not really fit figure 1a. The text says that the spun material is collected on the grounded collector. In the figure a turning cylindrical device is shown – how do you collect material on a turning cylindrical part? The photo of the real device in Figure 1b is rather small.

4. In addition, the name of PVDF must be corrected in line 36. The correct spelling is poly(vinylidene fluoride)! Otherwise the name of the polymer is polyvinylidene-certainly not correct.

5. Based on the IR band at 840 cm-1 it is concluded that PVDF occurs in its β phase. However, the gamma phase has a band at 840 cm-1, too. Therefore, assignment of the β phase requires additional information from XRD.

6. This brings me to the XRD image: Unfortunately, the XRD spectrum of the pure PVDF and of the composite material is too small to identify any peaks. With respect to the discussion on page 9, where it is stated in line 248/249 that the ZnO nanorods contribute to the formation of the all trans conformation (it should be mentioned that this is the β phase), it would be interesting to see whether the β phase peak in the XRD spectrum is increasing in going from the pure PVDF to the composite material.

7. Figure 4f: it needs to be stated to which sample the data presented refers. The Zn content appears to be rather small.

8. The English language requires careful revision. Not being a native speaker myself  I am careful with judging the English language, however, there are quite a number of phrases that are not o.k., at least to me.  Some examples:

Line 10 ... low voltage electrical voltage..

Line 28 … largely dependent on the ability of power sources…

Line 29 … The appearance of piezoelectric ….change to the occurrence of …?

Line 70/71 .. higher piezoelectric and mechanical properties … what’s the meaning of a higher mechanical property?

Line 105 ZnO nanorod (not nanorods) arrays

Line 105 … Teflon-lined stainless autoclave … change to stainless steel autoclave

Line 128 …with sputter coated with gold…

Line 137  A purpose-built device which consists of a piezoelectric device

Line 146…about 120 nm thin part and 230 nm thick one

Line 155 Figure 2d shows FTIR spectra not FTIR spectroscopy

Line 161 … are related to the bending vibration of the bending and the contraction vibration of CH2 (??)

Line 165 … bending peak of about CH.

Line 171… infrared spectrum curve

Line 185 … the braking tensile strength..

In addition, adverbs and adjectives are somewhat mixed up at several places.
